# Effects of word familiarity and receptive vocabulary size on speech-in-noise recognition among young adults with normal hearing

Meredith D. Braza[1,2]☺, Heather L. Porter[2‡], Emily Buss[1☺], Lauren Calandruccio[3‡], Ryan W. McCreery[2‡], Lori J. Leibold[2☺]*

1 Department of Otolaryngology/Head and Neck Surgery, The University of North Carolina at Chapel Hill, Chapel Hill, North Carolina, United States of America, 2 Center for Hearing Research, Boys Town National Research Hospital, Omaha, Nebraska, United States of America, 3 Department of Psychological Sciences, Case Western Reserve University, Cleveland, Ohio, United States of America

☺ These authors contributed equally to this work.
‡ HLP, LC and RWM also contributed equally to this work.
* lori.leibold@boystown.org

**Data Availability Statement:** De-identified data are found here: DOI 10.17605/OSF.IO/75UM2.

## Abstract

Having a large receptive vocabulary benefits speech-in-noise recognition for young children, though this is not always the case for older children or adults. These observations could indicate that effects of receptive vocabulary size on speech-in-noise recognition differ depending on familiarity of the target words, with effects observed only for more recently acquired and less frequent words. Two experiments were conducted to evaluate effects of vocabulary size on open-set speech-in-noise recognition for adults with normal hearing. Targets were words acquired at 4, 9, 12 and 15 years of age, and they were presented at signal-to-noise ratios (SNRs) of -5 and -7 dB. Percent correct scores tended to fall with increasing age of acquisition (AoA), with the caveat that performance at -7 dB SNR was better for words acquired at 9 years of age than earlier- or later-acquired words. Similar results were obtained whether the AoA of the target words was blocked or mixed across trials. Differences in word duration appear to account for nonmonotonic effects of AoA. For all conditions, a positive correlation was observed between recognition and vocabulary size irrespective of target word AoA, indicating that effects of vocabulary size are not limited to recently acquired words. This dataset does not support differential assessment of AoA, lexical frequency, and other stimulus features known to affect lexical access.

## Introduction

One challenge for understanding group differences in the ability to recognize speech in noise is that there are often substantial individual differences in performance even among young adults with normal hearing. Listeners can differ with respect to linguistic, cognitive, and perceptual abilities, resulting in differences in lexical access speed, verbal working memory,

**Funding:** This work was supported by a grant from the National Institute of Deafness and Other Communication Disorders (NIDCD) of the National Institutes of Health (NIH). R01 R01DC011038 (LJL). https://www.nidcd.nih.gov/ The funders had no role in study design, data collection and analysis, decision to publish, or preparation of the manuscript.

**Competing interests:** The authors have declared that no competing interests exist.

rhythm perception, inhibition, and vocabulary size [1, 2]. However, the influence of listener factors on speech-in-noise recognition appears to depend on characteristics of the stimuli used to evaluate performance. These stimulus characteristics include age of acquisition (AoA), lexical frequency, and phonotactic probability of the target speech materials, as well as signal-to-noise ratio (SNR) and masker type [3–6]. Beyond their contribution to accuracy, speech-related and listener-related factors also modulate the processing cost associated with perceiving speech in noise [7].

The relationship between a listener's receptive vocabulary knowledge and their ability to recognize speech in noise has been of particular interest to auditory researchers [3, 8–10]. An association between vocabulary size and speech-in-noise recognition has been observed for a range of target stimuli including meaningful sentences [11], semantically anomalous sentences [3, 12, 13], and isolated words [14]. A positive correlation between vocabulary size and speech recognition performance has been observed for children and adults who are hard of hearing [2, 15, 16], and for young school-age children with normal hearing [2, 15–17], this association is not always observed for adults [18] or for older school-age children with normal hearing [17].

Additional evidence supporting a link between receptive vocabulary and speech recognition in adverse listening conditions comes from studies investigating adults' understanding of speech produced by a talker with a speech disorder or with an unfamiliar dialect or accent [12, 19]. For example, Banks and colleagues evaluated sentence recognition in speech-shaped noise using an adaptive tracking procedure [19]. Participants were young adults with normal hearing who were native speakers of British English. Sentences produced by a native speaker of British English were digitally manipulated to create an unfamiliar accent. Vocabulary scores were negatively associated with speech recognition thresholds (SRTs); adults with larger vocabularies tended to have lower SRTs than adults with smaller vocabularies. Based on the results of these studies and those described above, it has been suggested that individual differences in language experience and/or lexical processing abilities contribute to the substantial variability in performance observed across listeners for a wide range of challenging listening conditions [9, 12, 19].

It has been posited that individual differences in knowledge of linguistic structure mediates the relationship between vocabulary size and speech-in-noise recognition [3, 20]. By this account, listeners vary in their implicit language knowledge (e.g., statistical properties of speech), which supports reconstruction of target speech that has been degraded by competing noise [21]. Support for this view was provided by Fletcher and colleagues, who showed a positive relationship between vocabulary size and speech-in-noise recognition at a range of SNRs in adults with normal hearing [3]. Of particular interest, the strength of this association varied nonmonotonically with SNR. The effect was largest at 1 dB SNR, corresponding to an average of approximately 67% correct recognition. A weaker association was observed at a more advantageous SNR of 4 dB (81% correct) and at less advantageous SNRs of -2 dB (45% correct) and -5 dB (20% correct). The authors suggested this nonmonotonic pattern of results reflected maximal benefit of implicit language knowledge when speech cues are present but moderately degraded by background noise.

The effect of receptive vocabulary on speech-in-noise recognition may also depend on the listener's familiarity with the target speech stimuli used to evaluate masked speech recognition abilities, where greater familiarity is associated with higher lexical frequency and earlier age of acquisition [14]. Specifically, listeners may require less acoustic-phonetic information to recognize familiar words than unfamiliar words. School-age children with larger vocabularies tend to perform better on speech-in-noise tasks than children with smaller vocabularies when target stimuli are later-acquired words, but this relationship is not always observed when the

target stimuli are early-acquired words [6, 17]. It is not clear whether this pattern of results extends to adults. When listening to speech in the presence of background noise, language demands for adults may be less than for children who are still developing early language skills. Early-acquired words tend to occur more frequently in spoken language relative to late-acquired words [22], and we know that lexical frequency affects recognition in both adults [3] and children [22]. Lexical frequency may therefore play a role in recognition of early- and later-acquired words.

Understanding the relationship between receptive vocabulary and speech-in-noise recognition when target words differ by AoA and lexical frequency may help us interpret individual differences in results from speech-in-noise testing obtained in research and clinical settings. The purpose of the present study was therefore to further our understanding of the effects of listener and stimulus factors on speech-in-noise recognition. Receptive vocabulary size and percent correct recognition of words presented in speech-shaped noise were estimated for adults with normal hearing. Performance was evaluated at two SNRs using words with an average AoA of 4, 9, 12, or 15 years. The motivation for manipulating SNR was to increase opportunities for observing effects of linguistic knowledge, which may vary with task difficulty [3]. For Experiment 1, word lists were mixed with respect to AoA. For Experiment 2, lists were either mixed or blocked by AoA to see if listener expectation regarding AoA affected performance. Experiment 3 evaluated data collected in the previous two experiments to evaluate possible effects of word duration.

Two main predictions were made. First, noise-masked word recognition was predicted to be better for early-acquired words than later-acquired words. Second, based on previous findings in children [6, 17], the strength of the association between receptive vocabulary size and percent-correct masked word recognition was predicted to be greater for later-acquired words than early-acquired words.

## General methods

### Participants

Participants were native speakers of American English between 19 and 50 years of age (mean = 31 years), all with self-reported normal hearing and no history of hearing loss. Recruitment was based on word of mouth and included people within social networks of lab staff. Each participant completed testing remotely in a single session lasting about 80 minutes, and none participated in more than one experiment. Participants were reimbursed $15/hour in electronic gift cards. All procedures were approved by the Boys Town National Research Hospital Institutional Review Board (IRB).

All participants self-reported that they had normal hearing. This remote study was conducted shortly after onset of the COVID-19 pandemic, when few valid remote screening instruments were readily available. Of the participants tested, 43% had documented normal hearing from recent participation in our in-lab studies and/or from being a test subject in the UNC Doctor of Audiology Program. Potential participants were excluded if they reported one or more ear infections in the past year or a history of hearing loss. While we cannot definitively rule out hearing loss in this cohort, concern over this possibility is tempered by the fact that effects of interest were evaluated within subjects.

### Remote testing procedures

For each experiment, participants completed two primary tasks: 1) receptive vocabulary testing, and 2) speech-in-noise testing. The order of these tasks was randomly assigned for each participant. Consent was obtained via a secure Webex video conference call. When it was time

for the participant to sign/date/fill out forms, the first author transferred keyboard and mouse control to the participant. Following the consent process, the participant received a remote testing kit which contained: 1) a tablet computer with a touchscreen and charger, 2) Sennheiser HD 25 –II headphones, 3) instructions including specification of test order for that participant, and 4) alcohol wipes. The kit was dropped off to the participant's door, and kits were cleaned with alcohol wipes between use. Instructions included strategies for ensuring a quiet test environment (e.g., turning off the TV, limiting distractions from family members and pets), when to take breaks, who to contact with questions, how to run each program, the steps for completing each task, and what to do when the tasks were completed. The first author was available to assist via videoconference if participants had questions as they were testing.

## Receptive vocabulary assessment

The Peabody Picture Vocabulary Test, Fourth Edition (PPVT-4) [23] was used to quantify receptive vocabulary. Stimuli were recorded in a sound booth by a 49-year-old male talker who is a native speaker of American English. Stimuli were presented via headphones at a comfortable listening level using a custom software program preinstalled on the tablet. In each trial, participants heard a word preceded by the carrier phrase, "Show me," and then identified the word by selecting the associated illustration on the touchscreen. Each trial was associated with four illustrations, scanned in from a hard copy version of the PPVT-4, including the correct response and three foils. Prior to beginning the task, participants completed a practice trial to ensure that they understood how to perform the task. Trial-by-trial data and summary scores were uploaded automatically to REDCap, a secure web app for storing and managing data [24], following completion of the task.

## Speech-in-noise testing

Target stimuli were drawn from a corpus of 240 disyllabic words, selected based on their AoA [25]. There were 60 words in each of four categories based on mean AoA: those acquired at 4 years of age (4.0–4.7 yrs, mean 4.4 yrs), 9 years of age (9.0–9.2 yrs, mean 9.1 yrs), 12 years of age (12.0–12.2 yrs, mean 12.1 yrs), and 15 years of age (14.4–15.9 yrs, mean 15.0 yrs). As expected, these four lists also differed with respect to lexical frequency. Fig 1 shows lexical frequency on a log scale, based on Brysbaert and New [26], plotted as a function of AoA for the four wordlists. One-tailed Welch's $t$-tests indicate that the log transformation of lexical frequency fell with increasing AoA across all four lists (4 vs. 9 yrs, $t(116) = 9.82$, $p < .001$; 9 vs. 12 yrs, $t(112) = 2.54$, $p = .006$; 12 vs. 15 yrs, $t(117) = 1.89$, $p = .030$). The four lists were balanced with respect to phonetic content. One- and two-phoneme probabilities were computed based on the Phonotactic Probability Calculator of Vitevitch and Luce [27]; those probabilities were not significantly different across lists ($p \leq .223$, uncorrected). Target words are reported in S1 Table.

Target words were produced by a 28-year-old male talker who is a native speaker of American English with no noticeable regional accent; productions were recorded inside a double-walled sound booth. Recordings were made with the talker's mouth positioned approximately 6 inches in front of a cardioid condenser microphone (Shure KSM 42 cardioid condenser). The talker was recorded saying the carrier phrase, "Say the word," prior to each target word. Recordings were made using a TwinFinity 710 preamplifier, M-Audio FastTrack Pro audio interface, and Logic-Pro-X recording software. Recorded words were then edited using Sound Studio to remove all silent periods before and after the recording. The individual files were scaled to equivalent root-mean-square level using MATLAB [28]. Target recordings ranged in duration from 1.2 to 2.4 seconds (mean of 1.8 sec), including the carrier phrase.

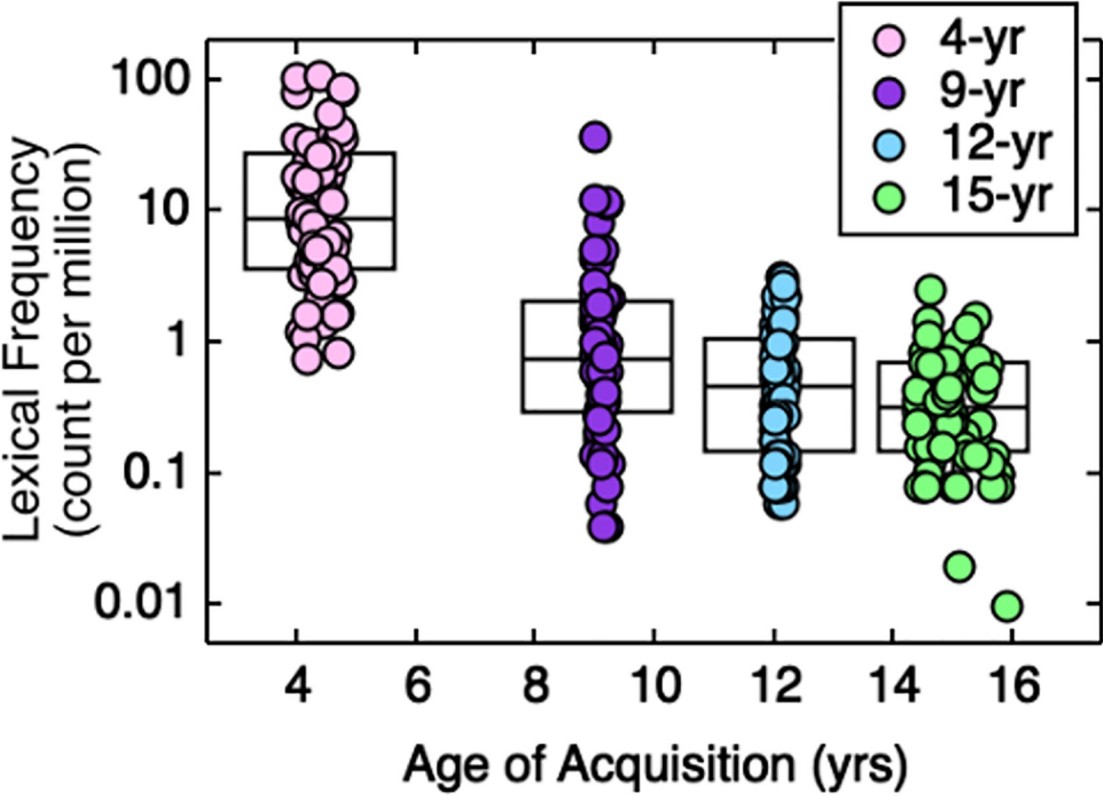

**Fig 1. Lexical frequency as a function of age of acquisition.** Symbol fill indicates the AoA category, as defined in the legend. Boxplots indicate the distribution of lexical frequency for each AoA category. Horizontal lines indicate the median, and boxes span the 25th to 75th percentiles.

The task was open-set word recognition in the presence of speech-shaped noise. Stimuli were presented diotically over headphones. The speech-shaped noise was generated based on the long-term average speech spectrum of the targets, including the carrier phrase. Custom software running on the tablet computer presented stimuli through the onboard soundcard and collected participants' responses. The target was temporally centered in a 3.4-second sample of masking noise, resulting in at least 500 ms of noise alone at the beginning and end of each stimulus presentation. The fidelity of sound from the onboard soundcard was evaluated by playing a test stimulus comprising a sequence of 1-sec pure tones, with 20-ms onset and off-set ramps. The first seven tones were at octave frequencies 125–8000 Hz, all presented at 75 dB SPL. The final four tones were at 1000 Hz, presented at 75, 65, 55, and 45 dB SPL. The output of the soundcard was routed to an oscilloscope. Visual inspection of the output did not reveal any frequency shaping or amplitude compression associated with the soundcard drivers.

Prior to testing, participants heard a passage produced by the target talker, and they were asked to adjust the volume to a comfortable listening level. Participants were explicitly instructed not to adjust the volume again for the remainder of the experiment. Upon equipment drop-off for each participant, volume on the tablet was set to 50%, which corresponded to approximately 75 dB SPL at the headphones, as measured using a 6-cc coupler and precision sound level meter (Larson Davis Model 824). Based on self-report, very few participants adjusted the volume from this pre-set level. Next, participants listened to instructions recorded by the target talker. The instructions described the task and directed the listener to ignore the

background noise and listen to the target talker. Participants were also instructed to repeat each word they heard aloud following each trial. Following each verbal response, participants were instructed to type their response into the response box. Data were automatically uploaded to REDCap [24].

## Data analysis

Linear regression models were used to evaluate logit-transformed percent correct data, and logistic regression was used to evaluate trial-by-trial responses. Scores on the PPVT and logit-transformed word scores were represented as z-scores to facilitate comparison of effect sizes. All models included a random intercept for each participant. *T*-tests were evaluated two-tailed unless otherwise indicated, with a significance criterion of $\alpha$ = .05. The Akaike Information Criterion (AIC) and Chi-square tests were used to compare alternative models.

## Experiment 1

Twenty-seven participants (9 males) completed Experiment 1. The average age of these participants was 32 years (20–49 yrs). For speech-in-noise testing, each participant heard four lists of 60 words. Each list contained an equal number of words from each AoA category (15 words with AoA of 4 yrs, 9 yrs, 12 yrs, and 15 yrs). Two lists were played at -5 dB SNR, and two lists were played at -7 dB SNR. The lists for each SNR and list order were randomly assigned for each participant.

### Results and discussion

Panels A1 and A2 of Fig 2 show the distribution of word recognition scores across participants for each AoA (Panel A1) and as a function of receptive vocabulary (PPVT score; Panel A2). Performance was better at -5 dB SNR than -7 dB SNR, and performance tended to be better for early-acquired words than later-acquired words. The one deviation from this trend was observed for early-acquired words presented at -7 dB SNR, where there was an unexpected trend for poorer performance for words with an AoA of 4 years than 9 years. These trends were evaluated with a linear regression model, with SNR and AoA represented as categorical factors. Participants' PPVT scores were included as a continuous variable. Results are reported in Table 1. There were significant effects of receptive vocabulary (*p* = .051) and SNR (*p* < .001). At -5 dB SNR, there was no difference between performance with AoA of 4 years compared to 9 years (*p* = .443), but there were differences compared to 12 and 15 years (*p* < .001). There was a significant interaction between SNR and AoA of 15 years (*p* = .026) and a non-significant trend for an interaction between SNR and AoA of 9 years (*p* = .068). Two-tailed paired *t*-tests for the -7 dB SNR data indicate that performance was *worse* for AoA of 4 years than 9 years (*p* < .001), *better* for 9 years than 12 years (*p* < .001), and not significantly different for 12 years and 15 years (*p* = .621). Interactions between receptive vocabulary and AoA were not significant (*p* $\geq$ .202). This result fails to support the hypothesis that effects of vocabulary size depend on AoA, although for later-acquired words there were greater mean beneficial effects of having a larger vocabulary.

One question remaining at the end of Experiment 1 was whether the trend for a nonmonotonic effect of AoA on word recognition at -7 dB SNR was a chance finding, or if this effect could be replicated. A nonmonotonic effect of AoA at -7 dB SNR was not anticipated at the outset, but one post-hoc explanation for this result has to do with listener expectation. Adults tested in this protocol might have prior expectations regarding the type of target words they were listening for, and those expectations could in turn affect performance, resulting in relatively better performance for words that conformed to those expectations as compared to

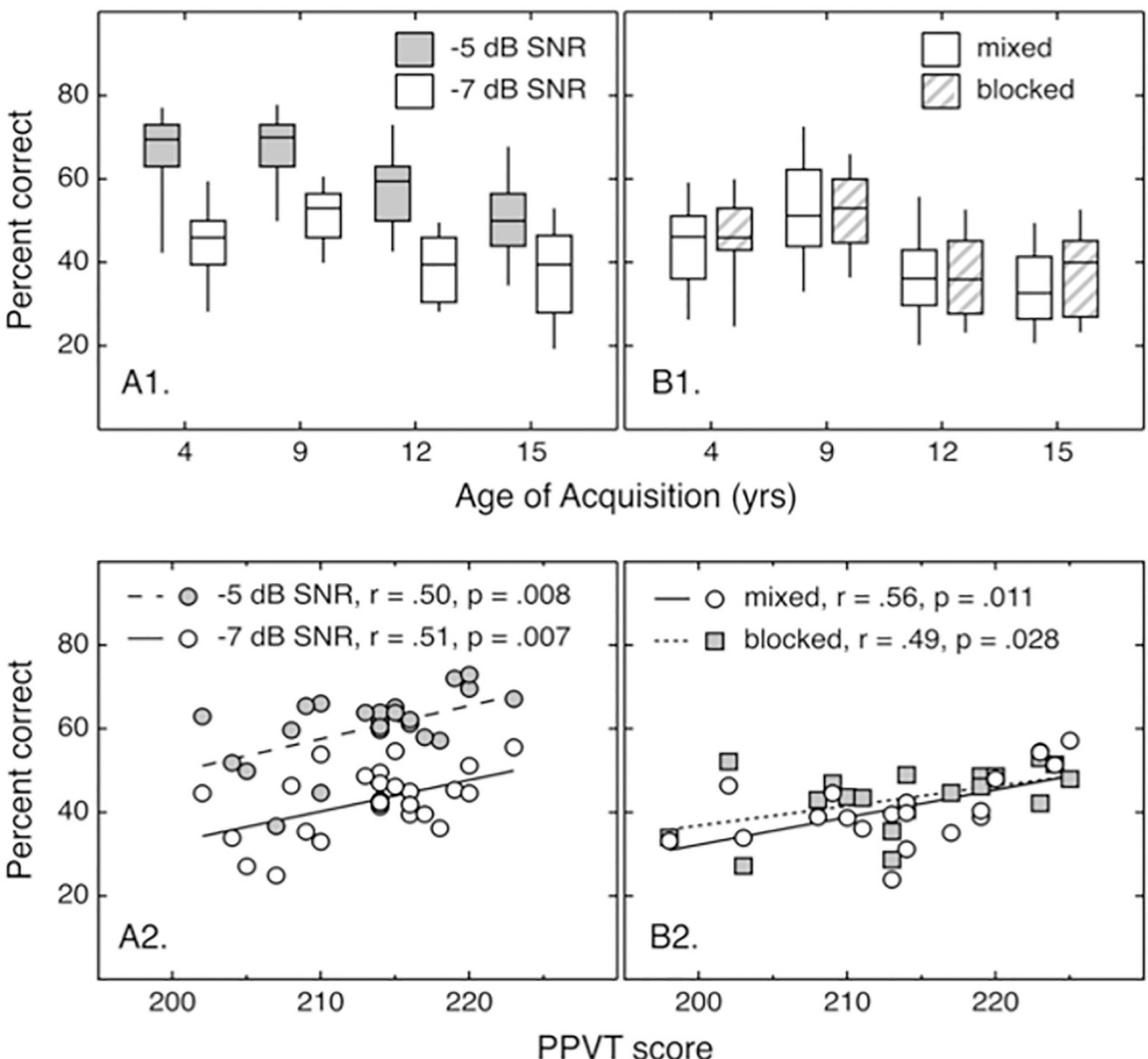

**Fig 2. Word scores by AoA and PPVT scores.** Panels A1 and B1 show the distribution of word scores, plotted as a function of AoA. The horizontal lines indicate the medians, boxes span the 25th to 75th percentiles, and whiskers span the 10th to 90th percentiles. Panels A2 and B2 show the mean SRT for individual listeners, plotted as a function of their receptive vocabulary (PPVT score). Lines indicate the association between percent correct and receptive vocabulary. Box and symbol fill reflects SNR (A1 & A2, Exp 1) or mixed vs. blocked wordlists (B1 & B2, Exp 2), as defined in the legend. All testing in Exp 2 was conducted at -7 dB SNR.

those that deviated from expectations. If participants were expecting to hear later-acquired words, this could result in relatively poor performance for words with a mean AoA of 4 years. It is not clear why listeners would form such an expectation, but the inclusion of a vocabulary assessment in the test protocol might be taken as evidence that this experiment was about recognition of more advanced vocabulary. This possibility was evaluated in Experiment 2 by providing participants with information about the types of target words to expect in select blocks of trials.

**Table 1. Linear mixed model for results of Experiment 1.**

|  | Value | SEM | DF | *t*-value | *p*-value |
|---|---|---|---|---|---|
| **(Intercept)** | 1.029 | 0.122 | 179 | 8.45 | < .001 |
| **PPVT** | 0.250 | 0.122 | 25 | 2.05 | .051 |
| **AoA(9yr)** | 0.107 | 0.139 | 179 | 0.77 | .443 |
| **AoA(12yr)** | -0.501 | 0.139 | 179 | -3.61 | < .001 |
| **AoA(15yr)** | -0.928 | 0.139 | 179 | -6.69 | < .001 |
| **SNR(-7)** | -1.336 | 0.139 | 179 | -9.63 | < .001 |
| **PPVT x AoA(9yr)** | 0.050 | 0.120 | 179 | 0.42 | .674 |
| **PPVT x AoA(12yr)** | 0.043 | 0.120 | 179 | 0.36 | .723 |
| **PPVT x AoA(15yr)** | 0.153 | 0.120 | 179 | 1.28 | .202 |
| **AoA(9yr) x SNR(-7)** | 0.360 | 0.196 | 179 | 1.84 | .068 |
| **AoA(12yr) x SNR(-7)** | 0.083 | 0.196 | 179 | 0.42 | .674 |
| **AoA(15yr) x SNR(-7)** | 0.440 | 0.196 | 179 | 2.24 | .026 |

This analysis includes fixed effects of PPVT score (*z*-score), AoA (reference = 4 yr), and SNR (reference = -5 dB). Interactions in the model included PPVT x AoA and AoA x SNR. Each row contains information about a parameter or the interaction between parameters, with factor levels indicated in parentheses.

## Experiment 2

Twenty participants (7 males) completed Experiment 2. The average age of these participants was 30 years (20–40 yrs). Participants heard six lists of words, all played at -7 dB SNR. The first two lists contained 60 words, each with an equal number of words from the four AoA categories, as in Experiment 1. The other four lists contained 30 words each and were blocked by AoA, such that each list contained words from only one of the AoA categories. Participants heard the two mixed lists first (Lists A and B, order randomized) followed by the four blocked lists (4, 9, 12, and 15 yrs AoA, order randomized). Prior to each blocked list, participants saw three words on the computer monitor exemplifying the AoA of the subsequent list; those example words had been previously considered for inclusion in the set of 240 targets but were not ultimately chosen for inclusion in that corpus. Other aspects of the stimuli and test procedures were the same as described for Experiment 1. Experiment 2 was conducted to determine whether the nonlinear effect of AoA at -7 dB SNR observed in Experiment 1 could be replicated with a new cohort of participants, and to evaluate the role of listener expectation on performance.

### Results and discussion

The right column of panels in Fig 2 shows the distribution of word scores across listeners for each AoA (Panel B1) and as a function of receptive vocabulary (PPVT score; Panel B2). Performance was similar for the mixed and blocked trials, and the effect of AoA was consistent with the -7 dB data from Experiment 1. This was confirmed with a linear regression model, with AoA and predictability of AoA (mixed vs. blocked) represented as categorical factors. PPVT scores were subjected to a z-transform and included as a continuous variable. Results appear in Table 2. There was a significant effect of receptive vocabulary (*p* = .008). The effect of AoA predictability (mixed vs. blocked) was not significant (*p* = .322). For the mixed condition, performance was significantly *worse* for AoA of 4 years than 9 years (*p* = .003), and significantly *better* for AoA of 4 years than either 12 years (*p* = .017) and 15 years (*p* = .001). There was no interaction between predictability and AoA (*p* ≥ .506). Given the lack of an effect of predictability, the mixed and blocked data were averaged, and the result was used to evaluate the effect

**Table 2. Linear mixed model for results of Experiment 2.**

|  | Value | SEM | DF | t-value | p-value |
|---|---|---|---|---|---|
| **(Intercept)** | -0.396 | 0.151 | 133 | -2.63 | .010 |
| **PPVT** | 0.219 | 0.073 | 18 | 3.00 | .008 |
| **Pred(Blocked)** | 0.184 | 0.185 | 133 | 1.00 | .322 |
| **AoA(9yr)** | 0.553 | 0.185 | 133 | 2.99 | .003 |
| **AoA(12yr)** | -0.447 | 0.185 | 133 | -2.42 | .017 |
| **AoA(15yr)** | -0.631 | 0.185 | 133 | -3.42 | .001 |
| **Pred(Blocked) x AoA(9yr)** | -0.174 | 0.261 | 133 | -0.67 | .506 |
| **Pred(Blocked) x AoA(12yr)** | -0.121 | 0.261 | 133 | -0.46 | .644 |
| **Pred(Blocked) x AoA(15yr)** | 0.085 | 0.261 | 133 | 0.33 | .744 |

This analysis includes fixed effects of PPVT score (z-score), AoA predictability (reference = mixed) and AoA (reference = 4 yr), as well as the interaction between predictability and AoA. Each row contains information about a parameter or the interaction between parameters, with factor levels indicated in parentheses.

of AoA via two-tailed paired *t*-tests. As observed in Experiment 1, performance was *worse* for an AoA of 4 years than 9 years ($p < .001$), *better* for an AoA of 9 years than 12 years ($p < .001$), and no difference was observed for an AoA of 12 years and 15 years ($p = .516$).

This experiment confirmed the nonmonotonic effect of AoA on word recognition for the -7 dB SNR presentation level for both fixed and blocked presentation, but it did not offer any possible explanation for that result. One consideration is whether the word lists used in the present set of experiments differed in ways that could have affected performance apart from differences in AoA. While the four word lists were balanced for one- and two-phoneme probabilities, there are many other factors that were not explicitly controlled and may have differed. Experiment 3 evaluated one such feature, target word duration.

## Experiment 3

The final experiment evaluated whether target word duration affects recognition for this stimulus set. Stimulus duration was evaluated by manually marking target word boundaries for each recording. The geometric mean of duration was 660 ms for an AoA of 4 years (IQR: 584–760 ms), 805 ms for an AoA of 9 years (IQR: 708–934 ms), 781 ms for an AoA of 12 years (IQR: 671–903), and 739 ms for an AoA of 15 years (IQR: 663–828 ms). Using uncorrected two-tailed t-tests, the 4-year words were significantly shorter than any of the other three AoA categories (p ≤ .009). The 9-year words were significantly longer than the 15-year words (p = .006) but not the 12-year words (p = .422). There was a non-significant trend for 12-year words to be longer than 15-year words (p = .060). Differences in duration across lists mirror the nonmonotonic pattern of performance as a function of AoA observed in the previous two experiments for the -7 dB SNR presentation level.

To evaluate the possible role of target word duration in performance, data were combined for the -7 dB SNR conditions in Experiment 1 and the mixed conditions of Experiment 2. This combined dataset included 49 participants. An analysis of trial-by-trial data was conducted using logistic regression, with random effects of subject and word. Fixed effects were the log transform of target word duration in ms, AoA (a categorical variable), PPVT (a continuous variable), and the interaction between PPVT and AoA. The results of this model are shown in Table 3. This model indicates a significant effect of PPVT (*p* = .044) and a significant effect of word duration (*p* = .019). Performance for words acquired at 4 and 9 years of age was not significantly different when word duration was included in the model (*p* = .507). This result is

**Table 3. Linear mixed model for combined data from Experiment 1 and 2, collected using -7 dB SNR level and mixed presentation.**

| | Value | SEM | *z*-value | *p*-value |
|---|---|---|---|---|
| **(Intercept)** | -6.897 | 2.767 | -2.493 | 0.013 |
| **PPVT** | 0.196 | 0.097 | 2.015 | 0.044 |
| **AoA(9yr)** | 0.263 | 0.397 | 0.663 | 0.507 |
| **AoA(12yr)** | -0.678 | 0.392 | -1.728 | 0.084 |
| **AoA(15yr)** | -0.738 | 0.384 | -1.921 | 0.055 |
| **Duration** | 1.546 | 0.657 | 2.353 | 0.019 |
| **PPVT x AoA(9yr)** | 0.116 | 0.105 | 1.100 | 0.272 |
| **PPVT x AoA(12yr)** | 0.115 | 0.107 | 1.081 | 0.280 |
| **PPVT x AoA(15yr)** | 0.069 | 0.106 | 0.651 | 0.515 |

This analysis includes fixed effects of target word duration in ms, PPVT score (*z*-score), AoA (reference = 4 yr), and the PPVT x AoA interaction. Each row contains information about a parameter or the interaction between parameters, with factor levels indicated in parentheses.

consistent with the idea that the longer duration of words acquired at 9 years of age could be responsible for the nonmonotonic performance as a function of AoA. In contrast to previous analyses, worsening performance for words acquired at 12 and 15 years of age approached, but did not reach significance in this analysis ($p \geq .055$). One caveat is that evaluating AoA as a categorical variable does not capture the ordered prediction associated with increasing AoA (e.g., that effects of AoA for 12-year words should be intermediate between 9-year and 15-year words). A second model with log of lexical frequency (a continuous variable) in place of AoA (a categorical variable) resulted in a reduction of the AIC and a non-significant change in model fit ($X^2(4) = 2.63$, $p = .620$). This result indicates that we cannot differentiate between effects of AoA and effects of lexical frequency in this dataset.

Evidence that differences in word duration are responsible for the nonmonotonic effect of AoA observed in Experiments 1 and 2 raises the question of whether these differences are particular to our stimulus set or whether they are representative of AoA-related differences in word duration inherent in the language. Text-to-speech synthesis was used to address this question, implemented in MATLAB [29]. Each word was synthesized separately. The resulting wav files included variable-duration segments of silence before and after the synthesized speech. To replicate the manual splicing used to evaluate stimulus recordings, the envelope was extracted via full-wave rectification and low-pass filtering twice with a 4$^{th}$ order 40-Hz Butterworth (once forward and once backward). The first and last time point that was $\geq 40$ dB down from the peak was used to define the beginning and end of the word, respectively.

The first step was to confirm that synthesized speech replicated the mean differences in list duration for the target words. The geometric mean of durations by list were 674 ms for an AoA of 4 years (IQR: 564–794 ms), 767 ms for an AoA of 9 years (IQR: 718–864 ms), 710 ms for an AoA of 12 years (IQR: 629–816 ms), and 697 ms for an AoA of 15 years (IQR: 621–781 ms). Across lists, the duration estimated using the text-to-speech algorithm was 4.6% greater than the duration of the recorded stimuli, and the correlation between these values was r = .63 (p < .001). These results suggest that the differences in duration as a function of AoA for the recorded speech stimuli are due in part to phonetic features of the target words, and that text-to-speech synthesis can be used to characterize these differences.

The second step was to estimate duration as a function of AoA for a larger set of two syllable words in the Kuperman database. An open-source dictionary maintained by Carnegie Mellon

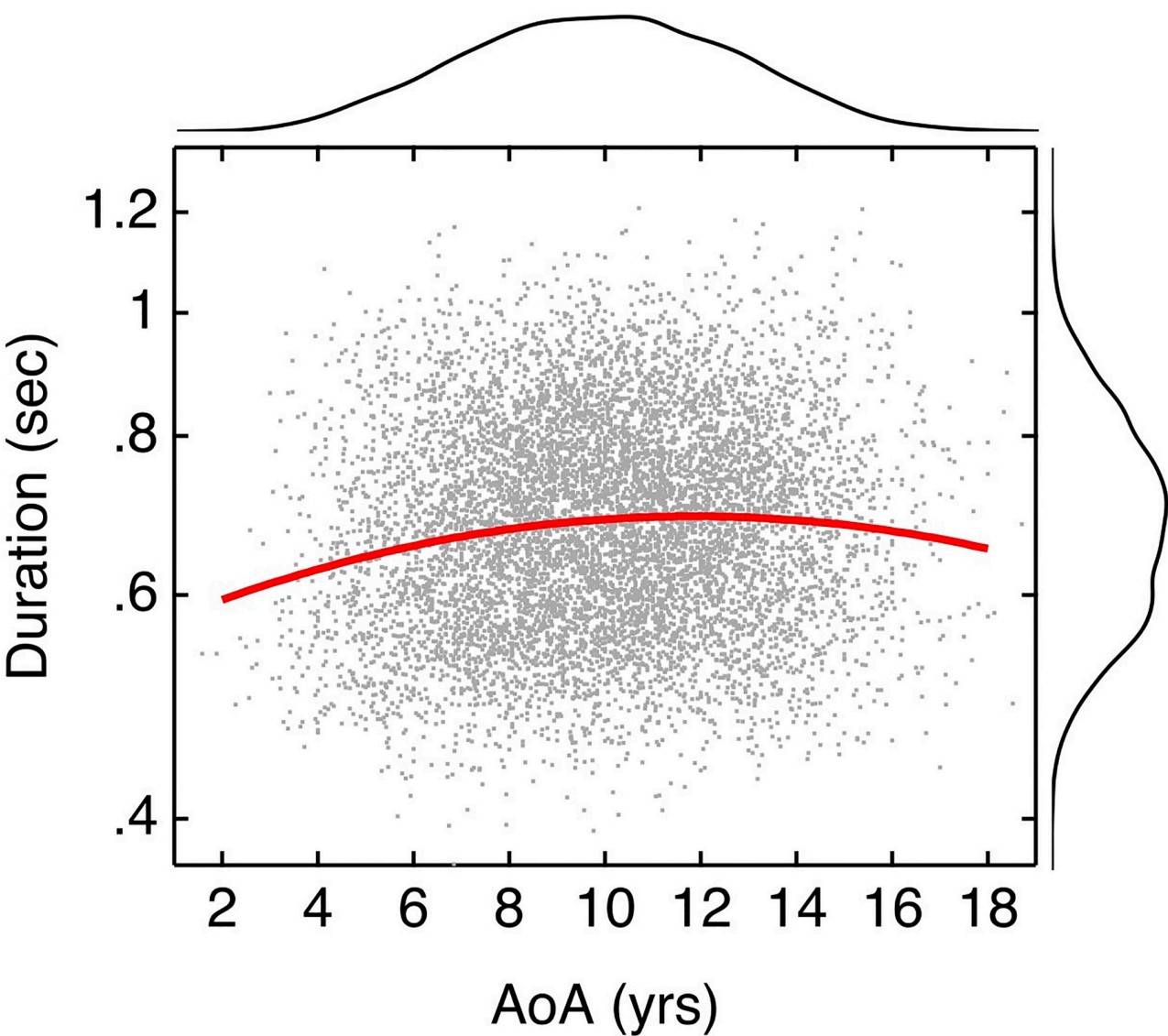

**Fig 3. Word duration in sec as a function of AoA.** Word duration was estimated using text-to-speech synthesis for two-syllable words from the Kuperman database. Those estimates of duration in seconds are plotted as a function of AoA in years, shown with grey dots. A three-parameter line fit is shown in red, indicating modest increase in duration with increasing AoA up to 12 years, followed by modest reductions in duration thereafter. The distributions of duration and AoA are indicated in the right and top margins, respectively.

University (CMUDict [30]) was used to define the number of syllables. That corpus contains 133,779 words, of which 61,468 are pronounced as disyllabic. Cross-referencing the two-syllable words from the CMUDict database with the Kuperman database identified 8,381 two-syllable words. These words were synthesized, and duration was estimated using the methods described above. Fig 3 shows estimates of word duration plotted as a function of AoA, with distributions of both parameters indicated in the margins. The solid red line indicates a three-parameter polynomial fit to the natural log transform of duration ($y = \text{-}1.553\text{e-}3 \cdot x^2 + 3.684\text{e-}2 \cdot x - 5.867\text{e-}1$). All three of these parameters were significantly different from zero. This function predicts increases in duration for words with an AoA up to ~12 yrs, and a modest trend for decreasing duration with further increases in AoA. For the AoA categories in the present

experiment, this corresponds to a 55-ms increase in duration between AoAs of 4 to 9 years and a 2-ms decrease between AoAs of 9 and 15 years. These changes are smaller than those observed in recorded stimuli (145 ms increase and 66 ms decrease, respectively), suggesting that the magnitude of differences in word duration for stimuli used in Experiments 1 and 2 may not be representative of all two-syllable words with the associated AoAs.

## General discussion

The purpose of this study was to evaluate effects of word familiarity, as indexed by AoA, and individual differences in receptive vocabulary on word recognition in noise for young adults with normal hearing. There were two a priori predictions. The first prediction was that participants would perform more poorly on the speech-in-noise task for later-acquired words with a lower lexical frequency than for early-acquired words with a higher lexical frequency. The second prediction was that the strength of the association between vocabulary size and speech-in-noise recognition scores would depend on word familiarity, with larger effects for less familiar words (i.e., those with later AoA and lower lexical frequency). Target word AoA and lexical frequency tend to be correlated ([31]), and that association was clear in the current stimulus set. Therefore, whereas the word sets were selected based on AoA, and are described as such in the discussion that follows, it is not clear whether AoA, lexical frequency, word duration, or a combination of those factors are responsible for the effects observed.

The prediction that word recognition in noise would be better for early-acquired words than late-acquired words was evaluated by comparing percent-correct speech recognition scores at two different SNRs for words with an average AoA of 4, 9, 12, or 15 years. The findings were in partial agreement with this prediction. Overall word recognition performance was better at -5 dB relative to -7 dB SNR, and percent-correct scores tended to decrease as AoA increased. This pattern of results is consistent with prior findings showing an association between word familiarity and masked speech recognition [3, 14, 32, 33]. For example, Savin [31] estimated speech-in-noise thresholds in a group of young adults using words that spanned a wide range with respect to frequency of occurrence and word length [33]. For relatively short words (e.g., monosyllabic and disyllabic words), SRTs were lower for frequently occurring words than infrequently occurring words.

Although there was a general trend for speech-in-noise scores to decrease with increasing AoA, better performance for words with an average AoA of 9 years than 4 years was observed at -7 dB SNR. This pattern of results was not observed at the more advantageous SNR of -5 dB; similar speech-in-noise performance was observed for words acquired at 4 and 9 years at -5 dB SNR. The nonmonotonic effect of AoA at -7 dB SNR was observed for mixed and fixed blocks, inconsistent with the idea that listener expectation was responsible for the higher masked speech recognition scores observed for the 9- versus 4-year-old words. Recall that Fletcher and colleagues also observed a level effect in the relationship between AoA and word recognition [3]. However, the details of that effect were not the same as observed in the present study. Whereas Fletcher and colleagues observed a larger effect of AoA at a moderate SNR (corresponding to ~67% correct performance), we observed different effects for words acquired at 4 and 9 years of age despite similar percent correct scores at -5 dB SNR.

One potential explanation for the unexpected difference in performance between words acquired at 4- and 9-years at the more challenge SNR is that that the 9-year-old words were longer than the 4-year-old words. An analysis of target word duration determined that words acquired at 4 years of age tended to be shorter than those acquired later, and words acquired at 9 years tended to be longer than those acquired at 15 years of age. Including word duration in a statistical model of trial-by-trial responses indicates no significant difference between

recognition of words acquired at 4 years and 9 years of age. This suggests that the nonmonotonic effect of AoA observed in mean data could be due to differences in word duration across lists. Better recognition of longer words could be related to prior data indicating better performance for slower speaking rates and for words with more syllables [33, 34]. Analysis of a larger set of two-syllable words suggests that there is a nonmonotonic pattern of word duration as a function of AoA in English, but that the magnitude of duration differences observed with the stimuli used for Experiments 1 and 2 is larger than expected based on this analysis.

A major goal of this study was to evaluate the relationship between receptive vocabulary and masked speech recognition, considering the AoA of target words used to assess speech-in-noise performance. We anticipated a greater positive correlation between receptive vocabulary and percent-correct word recognition scores for later-acquired words relative to early-acquired words. This prediction was based on prior studies investigating adults' speech perception abilities under adverse listening conditions [3, 12, 19], and on studies of masked speech recognition in both children with normal hearing and children who are hard of hearing [6, 17]. In these studies, target stimuli were selected to fall within the lexicon of the youngest listeners. Thus, one explanation offered for the lack of an association between receptive vocabulary size and masked speech recognition performance for older children and adults in the earlier studies was that having a large receptive vocabulary is less beneficial for speech-in-noise recognition when using highly familiar target speech [6, 17].

In contrast to our initial prediction, there was a comparable positive relationship between receptive vocabulary size and masked word recognition performance for all four AoA categories and for both SNRs examined. While unexpected, these results are in agreement with findings from previous studies in which young adults with larger vocabularies showed an advantage when listening in noise relative to young adults with smaller vocabularies [2, 13]. One implication of these results is that limiting the test corpus to early-acquired words may not reduce effects of linguistic knowledge, as is often assumed when testing speech perception in a clinical setting. Whereas the present study found a consistent effect of vocabulary size on young adults' recognition of early- and later-acquired target words, several previous studies have reported effects of vocabulary size only for later-acquired targets [6, 17]. One potential explanation for discrepancies observed across studies was suggested over 50 years ago by Savin [33], who examined effects of lexical frequency on adults' word recognition in noise using words that varied in length from 1 to 8 syllables. While a strong association between lexical frequency and speech-in-noise performance was observed for short words, almost no effect of lexical frequency was observed for longer words. Savin posited that listeners hear each syllable in a long word in the context of the other syllables, improving speech recognition performance in a manner similar to that observed for words in semantically meaningful sentences [35]. Most previous studies that failed to show a consistent association between receptive vocabulary size and speech-in-noise recognition used sentence-length materials [6, 17], which might tend to underestimate the effects of vocabulary size because of the availability of sentence-level semantic and syntactic cues.

The current study had several limitations that could be addressed in future studies. Testing participants with wider variability in receptive vocabulary size could help generalize the role of receptive vocabulary on speech-in-noise performance to a broader population of listeners. Participants in this study had mean PPVT scores ranging from 198 to 225 (mean = 214), corresponding to the 30th to 99th percentiles for this age group (mean = 72.5th percentile). Measuring percent correct word recognition based on keywords in sentences rather than single words could shed light on the conditions under which effects of receptive vocabulary are observed. Finally, the present study focused on word familiarity, as indexed by AoA, but it is unclear whether familiarity is the dominant factor responsible for better recognition of early-

acquired words. There is a rich literature in psycholinguistics on the differential contributions of plasticity, cumulative word frequency, and semantic structure of lexical representation (reviewed by [36]), as well as effects of phonetic similarity across words and semantic richness [31, 37]. It is likely that selecting test words based on AoA affects results via multiple factors [38]. More detailed characterization of the test stimuli in future studies of masked speech recognition could help clarify the differential contributions of lexical neighborhood density, imageability, semantic richness, current and cumulative frequency, and AoA.

The results of the present study contribute to the growing body of evidence that remote testing is a feasible method of gathering data on speech recognition [39–41]. Laboratory hardware was delivered to listeners' homes, limiting inconsistencies related to the use of personal computers and headphones [39]. Remote testing options are appealing beyond the COVID-19 pandemic, particularly for those without access to traditional laboratory environments and/or when recruiting participants across a wide geographical region.

One implication of this study is that word familiarity, as indexed by AoA, is an important consideration when creating and interpreting clinical tests of masked word recognition. This study demonstrates that even for young adults with relatively large vocabularies, familiar words (those acquired early and/or with higher lexical frequency) are easier to recognize than less familiar words (those acquired later and/or with lower lexical frequency). After controlling for differences in word duration, there was monotonic trend for poorer performance with increasing AoA. There also may be differing effects of AoA based on the SNR of the stimuli. As such, it may be useful to test masked word recognition using a range of target word AoAs and SNRs to get a full picture of how these factors influence speech recognition ability. Considering these individual differences can also help clinicians and researchers more accurately interpret speech-in-noise recognition results.

## Supporting information

**S1 Table. Target words in the four AoA categories.** Disyllabic target words acquired at 4 years, 9 years, 12 years, and 15 years of age.
(DOCX)

## Acknowledgments

The authors would like to thank all the listeners who participated in the study and the members of the Human Auditory Development Laboratory at Boys Town National Research Hospital, specifically Margaret Miller and Raj Persaud for their assistance with data collection and programming and Tiana Cowan for her assistance with manuscript review.

## Author Contributions

**Conceptualization:** Meredith D. Braza, Heather L. Porter, Emily Buss, Ryan W. McCreery, Lori J. Leibold.

**Data curation:** Meredith D. Braza.

**Formal analysis:** Emily Buss, Lori J. Leibold.

**Funding acquisition:** Lori J. Leibold.

**Investigation:** Meredith D. Braza.

**Methodology:** Meredith D. Braza, Emily Buss, Lauren Calandruccio, Lori J. Leibold.

**Project administration:** Meredith D. Braza, Heather L. Porter.

**Supervision:** Emily Buss, Lori J. Leibold.

**Validation:** Meredith D. Braza, Lori J. Leibold.

**Writing – original draft:** Meredith D. Braza.

**Writing – review & editing:** Meredith D. Braza, Heather L. Porter, Emily Buss, Lauren Calandruccio, Ryan W. McCreery, Lori J. Leibold.

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
