## [Decision Letter · Decision Letter 0]

9 Dec 2021

PONE-D-21-35370Effects of Word Familiarity and Receptive Vocabulary Size on Speech-in-Noise Recognition Among Young Adults with Normal HearingPLOS ONE

Dear Dr. Leibold,

Thank you for submitting your manuscript to PLOS ONE. I have reviews from one expert in the field, but have  been unable to find another scientist with the required expertise to review your manuscript. I have read it myself, and decided to make a decision with the input of the one reviewer and my own assessment. It is my opinion that it would be highly unlikely that input from another reviewer would lead to a different decision, and I don't want you to have to wait any longer. Both the reviewer and I concluded that your study was conducted in largely an appropriate manner and the manuscript itself is well written. Nonetheless, we do have recommendations for you to consider in a revision. Therefore, I invite you to submit a revised version of the manuscript after you carefully consider the points raised during the review process and make the changes that you deem appropriate.

We look forward to receiving your revised manuscript.

Kind regards,

Susan Nittrouer, Ph.D.

Academic Editor

PLOS ONE

Journal Requirements:

Additional Editor Comments:

This was a reasonable study. The Introduction was especially well written.

The purpose was to examine whether word familiarity and receptive vocabulary size affect speech recognition thresholds for young adults with normal hearing listening to isolated words.

Although the study is fine as it is, I had two concerns, both related to the facts that I was not especially fond of the way word familiarity or vocabulary size were assessed and benchmarked. Age of acquisition is an extremely broad term and might not apply to each specific listener in the same manner; instead, age of acquisition may differ across individuals. The PPVT is a relatively insensitive measure of vocabulary size because one need only to have a low level of familiarity with a word in order to be able to select the picture that represents that word. Even measures of expressive vocabulary tap into a slightly deeper level of vocabulary knowledge than receptive vocabulary measures because the subject must retrieve the word from his/her own lexicon. David Pisoni developed a task that derives a more sensitive measure of the depth of a subject’s lexical knowledge; it is the Word Fam task. I highly recommend that task for future work.

I also had a slightly different interpretation of the (weak) age-of-acquisition effects that were found. Early-acquired words – which are high frequency, as the authors suggest – may not require listeners to recover phonological structure to the same extent as later-acquired, more ‘difficult’ words. Therefore, less may be needed in terms of acoustic-phonetic information in order to recognize early acquired words. That idea might be important to interpretation of the data obtained in this study, especially if we entertain another notion often invoked when discussing speech-in-noise recognition, and that is the idea of listening in the dips. Although speech-shaped noise has few temporal or spectral dips, it does have some. Later-acquired, less-familiar words may require the listener to recover more “bits” of signal information from across the spectrum and temporal structure than earlier-acquired, more-familiar words. And listeners’ abilities to recover those signal bits and integrate them might help explain individual differences in speech-in-noise recognition.

My final comment was that I found the Table 1 difficult to understand. It needs to be revised so that readers can immediately recognize what each row represents.

Reviewers' comments:

Reviewer's Responses to Questions

**Comments to the Author**

1. Is the manuscript technically sound, and do the data support the conclusions?

Reviewer #1: Yes

2. Has the statistical analysis been performed appropriately and rigorously? 

Reviewer #1: Yes

3. Have the authors made all data underlying the findings in their manuscript fully available?

Reviewer #1: Yes

4. Is the manuscript presented in an intelligible fashion and written in standard English?

Reviewer #1: Yes

5. Review Comments to the Author

Reviewer #1: The manuscript describes the results of studies of the effects of vocabulary size on open-set speech-in-noise recognition for adults with normal hearing. Data were collected using relatively standard methods, adapted to reflect Covid 19 restrictions. The general finding was that larger vocabulary size was associated with better recognition of speech in noise, consistent with some previous reports from others. A hypothesis that recognition of speech in noise would be associated with age of acquisition (or lexical frequency, with which it is correlated) was not supported. Because of Covid 19, testing was done in the listeners’ home using tablet computers equipped for that purpose. Some sort of accommodation was unavoidable, and that was a reasonable choice that is preferable in some ways to alternatives such as testing online. However, it is not a perfect solution, and I’ve mentioned a few issues in the Specific Comments.

Specific comments:

Line 122: It would be preferable in a study like this to confirm that the listeners had normal hearing. It seems likely that the authors relied on self-report of hearing status because testing was done remotely, but it would not have been difficult to include a simple pure-tone screening app on the tablet used for the actual study so that’s not a convincing explanation. However, screening would have required better control of absolute presentation level, which listeners were “asked to adjust” (line 197). This situation cannot be changed, but perhaps a brief justification for the choices that were made could be provided.

Line 174: Was this talker the same one who recorded the vocabulary words? I don’t think that would affect the outcome, but some readers might want to know.

Line 176, “The talker stood approximately 6 inches in front …”: Presumably this was intended to say that the talker’s mouth was 6 inches from the microphone?

Line 194-195, “Visual inspection of the [soundcard] output did not reveal any frequency shaping or amplitude compression associated with the soundcard drivers.” This information would be more useful if the measurements had been made on the signals after they’d been transduced by the earphone; that’s where frequency shaping is most likely to be introduced. Even if those measurements were not made prior to testing, presumably they could be made after the fact.

6. PLOS authors have the option to publish the peer review history of their article (what does this mean?). If published, this will include your full peer review and any attached files.

Reviewer #1: No

---

## [Author Response · Author response to Decision Letter 0]

9 Feb 2022

Manuscript PONE-D-21-35370

Response to Reviewers

Susan Nittrouer

Academic Editor

PLOS ONE

 February 9, 2022

Dear Dr. Nittrouer,

Thank you for your consideration of our paper entitled “Effects of Word Familiarity and Receptive Vocabulary Size on Speech-in-Noise Recognition Among Young Adults with Normal Hearing” (PONE-D-21-35370) for PLOS ONE, and for enclosing your comments. 

We have carefully reviewed the comments and have revised the manuscript accordingly. The big issues we addressed were 1) justifying the choice to base participant hearing status on self-report, 2) clarifying some of the signal recording methods, and 3) and improving the readability of Tables 1 and 2. In addition to these changes, we have added an analysis of word duration, which appears to account for the nonmonotonic effect of word duration that we observed behaviorally. 

Our responses to the reviewer comments are given point-by-point below and our changes to the manuscript are coded in red font. We believe the revised version of the manuscript has improved based on feedback and hope that you now find it suitable for publication. 

Best,

Meredith D. Braza

Doctor of Audiology Student

The University of North Carolina at Chapel Hill

 

Response to Reviewer 1:

RESP: Thank you for your review of our paper. We have responded to each of your points below.

The manuscript describes the results of studies of the effects of vocabulary size on open-set speech-in-noise recognition for adults with normal hearing. Data were collected using relatively standard methods, adapted to reflect Covid 19 restrictions. The general finding was that larger vocabulary size was associated with better recognition of speech in noise, consistent with some previous reports from others. A hypothesis that recognition of speech in noise would be associated with age of acquisition (or lexical frequency, with which it is correlated) was not supported. Because of Covid 19, testing was done in the listeners’ home using tablet computers equipped for that purpose. Some sort of accommodation was unavoidable, and that was a reasonable choice that is preferable in some ways to alternatives such as testing online. However, it is not a perfect solution, and I’ve mentioned a few issues in the Specific Comments.

Specific comments

Line 122: It would be preferable in a study like this to confirm that the listeners had normal hearing. It seems likely that the authors relied on self-report of hearing status because testing was done remotely, but it would not have been difficult to include a simple pure-tone screening app on the tablet used for the actual study so that’s not a convincing explanation. However, screening would have required better control of absolute presentation level, which listeners were “asked to adjust” (line 197). This situation cannot be changed, but perhaps a brief justification for the choices that were made could be provided.

RESP: To move forward with the consenting process, participants had to first verify that they do not believe they have hearing loss and have not had any ear infections in the past year. There is precedent in the psycholinguistics literature for using self-report as an indication of hearing status while doing supra-threshold speech perception experiments.

As recruitment was based on word of mouth and included people within social networks of lab staff, many participants had documented normal hearing from participating in our labs previously and/or from being a test subject in the UNC Doctor of Audiology Program. Of the 47 participants in our study, 20 (43%) had normal hearing documented (13 participants in Experiment 1, 7 participants in Experiment 2). Concern over possible hearing loss the remaining participants is tempered by the fact that the main questions of interest in this study were based on comparison of results across conditions. This has been added to the manuscript. 

Line 174: Was this talker the same one who recorded the vocabulary words? I don’t think that would affect the outcome, but some readers might want to know.

RESP: The talker who recorded the target words also recorded the carrier phrase, but a different talker recorded the PPVT words. This has been clarified in the manuscript by specifying each talker’s age. 

Line 176, “The talker stood approximately 6 inches in front …”: Presumably this was intended to say that the talker’s mouth was 6 inches from the microphone?

RESP: Yes, the talker’s mouth was 6 inches from the microphone. This has been clarified in the manuscript. 

Line 194-195, “Visual inspection of the [soundcard] output did not reveal any frequency shaping or amplitude compression associated with the soundcard drivers.” This information would be more useful if the measurements had been made on the signals after they’d been transduced by the earphone; that’s where frequency shaping is most likely to be introduced. Even if those measurements were not made prior to testing, presumably they could be made after the fact.

RESP: The goal of these measurements was to evaluate whether the soundcard was applying any signal conditioning. Many commercially available soundcard drivers modify the spectral and/or amplitude characteristics of the sound in their default configuration (e.g., bass boost or compression). Since this study relied on the soundcard of a tablet, we wanted to confirm that these features were not affecting the signal. Demonstrating linearity of the soundcard allows readers to rely on the frequency response published by the headphone manufacturer.

Response to Academic Editor:

RESP: Thank you for your review of our paper. We have responded to each of your points below.

This was a reasonable study. The Introduction was especially well written.

The purpose was to examine whether word familiarity and receptive vocabulary size affect speech recognition thresholds for young adults with normal hearing listening to isolated words.

Although the study is fine as it is, I had two concerns, both related to the facts that I was not especially fond of the way word familiarity or vocabulary size were assessed and benchmarked. Age of acquisition is an extremely broad term and might not apply to each specific listener in the same manner; instead, age of acquisition may differ across individuals. The PPVT is a relatively insensitive measure of vocabulary size because one need only to have a low level of familiarity with a word in order to be able to select the picture that represents that word. Even measures of expressive vocabulary tap into a slightly deeper level of vocabulary knowledge than receptive vocabulary measures because the subject must retrieve the word from his/her own lexicon. David Pisoni developed a task that derives a more sensitive measure of the depth of a subject’s lexical knowledge; it is the Word Fam task. I highly recommend that task for future work.

RESP: Thank you for the suggestion to consider using the task developed by David Pisoni in our future work. We elected to utilize the PPVT because 1) it is shown to be associated with speech perception outcomes in previous research, and 2) it could be easily adapted to a tablet-based format that was straightforward for remote participants. 

I also had a slightly different interpretation of the (weak) age-of-acquisition effects that were found. Early-acquired words – which are high frequency, as the authors suggest – may not require listeners to recover phonological structure to the same extent as later-acquired, more ‘difficult’ words. Therefore, less may be needed in terms of acoustic-phonetic information in order to recognize early acquired words. That idea might be important to interpretation of the data obtained in this study, especially if we entertain another notion often invoked when discussing speech-in-noise recognition, and that is the idea of listening in the dips. Although speech-shaped noise has few temporal or spectral dips, it does have some. Later-acquired, less-familiar words may require the listener to recover more “bits” of signal information from across the spectrum and temporal structure than earlier-acquired, more-familiar words. And listeners’ abilities to recover those signal bits and integrate them might help explain individual differences in speech-in-noise recognition.

RESP: An association between age of acquisition and the minimum cues required for recognition does seem like a reasonable interpretation of our results. While dips in nominally steady noise could theoretically be associated with performance, variability in target level would likely be the dominant factor in variable audibility over time. Additional discussion of the association between age of acquisition and the number/quality of cues required for recognition has been added to the text.

My final comment was that I found the Table 1 difficult to understand. It needs to be revised so that readers can immediately recognize what each row represents.

RESP: Each row contains information about a parameter or the interaction between parameters, with factor levels indicated in parentheses. This has been indicated in the legend. The same revisions were made to Table 2.

---

## [Editor Report · Decision Letter 1]

14 Feb 2022

Effects of Word Familiarity and Receptive Vocabulary Size on Speech-in-Noise Recognition Among Young Adults with Normal Hearing

PONE-D-21-35370R1

Dear Dr. Leibold,

Thank you for submitting your revised manuscript to PLOS ONE, and for your careful attention to the comments of the reviewer. At this time we are pleased to inform you that your manuscript has been judged scientifically suitable for publication and will be formally accepted for publication once it meets all outstanding technical requirements.

Kind regards,

Susan Nittrouer, Ph.D.

Academic Editor

PLOS ONE

---

## [Editor Report · Acceptance letter]

2 Mar 2022

PONE-D-21-35370R1 

Effects of Word Familiarity and Receptive Vocabulary Size on Speech-in-Noise Recognition Among Young Adults with Normal Hearing 

Dear Dr. Leibold:

I'm pleased to inform you that your manuscript has been deemed suitable for publication in PLOS ONE. Congratulations! Your manuscript is now with our production department. 

Kind regards, 

on behalf of

Dr. Susan Nittrouer 

Academic Editor

PLOS ONE